# Are We Prepared for the Next Pandemic? Management, Systematic Evaluation and Lessons Learned from an In-Hospital COVID-19 Vaccination Centre for Healthcare Workers

**DOI:** 10.3390/ijerph192316326

**Published:** 2022-12-06

**Authors:** Ana Zhelyazkova, Kristina Adorjan, Selina Kim, Matthias Klein, Stephan Prueckner, Philipp Kressirer, Alexander Choukér, Michaela Coenen, Sophia Horster

**Affiliations:** 1Institute of Emergency Medicine and Management in Medicine, University Hospital, Ludwig Maximilian University of Munich, 80336 Munich, Germany; 2Department of Psychiatry and Psychotherapy, University Hospital Munich, Ludwig Maximilian University of Munich, 80336 Munich, Germany; 3Department of Neurology, University Hospital Munich, Ludwig Maximilian University of Munich, 81377 Munich, Germany; 4Department of Communication and Media, University Hospital Munich, Ludwig Maximilian University of Munich, 80336 Munich, Germany; 5Laboratory of Translational Research Stress and Immunity, Department of Anaesthesiology, University Hospital Munich, Ludwig Maximilian University of Munich, 81377 Munich, Germany; 6Institute for Medical Information Processing, Biometry, and Epidemiology—IBE, Chair of Public Health and Health Services Research, Ludwig Maximilian University of Munich, 81377 Munich, Germany; 7Pettenkofer School of Public Health, 81377 Munich, Germany; 8Department of Gastroenterology and Hepatology, University Hospital Munich, Ludwig Maximilian University of Munich, 81377 Munich, Germany; 9Department of Commercial Directorate, University Hospital Munich, Ludwig Maximilian University of Munich, 81377 Munich, Germany

**Keywords:** COVID-19, vaccination centre, healthcare workers, occupational health

## Abstract

Background: the organisation of a COVID-19 vaccination campaign for healthcare workers (HCWs) within a university hospital presents a challenge of a particularly large scale and urgency. Here, we evaluate the in-hospital vaccination process and centre for HCWs at LMU University Hospital in Munich, Germany. Methods: We executed a mixed-method process evaluation of the vaccination centre at LMU University Hospital during the first COVID-19 vaccination campaign. In a programme monitoring, we continuously assessed the implementation of the centre’s operational management including personnel resources. In evaluating the outreach to and satisfaction of the target group with the centre and process, we executed two anonymous surveys aimed at the HCWs vaccinated at the in-hospital centre (1) as well as centre staff members (2). Results: staff numbers and process time per person were reduced several times during the first vaccination campaign. Lessons concerning appointment scheduling were learned. HCWs vaccinated at the in-hospital centre were satisfied with the process. A longer waiting time between admission and inoculation, perceived dissatisfying accessibility as well as an increased frequency of observed adverse events were linked to a reduced satisfaction. Comparatively subpar willingness to adhere to non-pharmaceutical measures was observed. Centre staff reported high satisfaction and a workload relatively equal to that of their regular jobs. Our outcomes provide references for the implementation of an in-hospital vaccination centre in similar settings.

## 1. Introduction

Vaccinations are among the most effective preventive measures against COVID-19 [1,2]. Once the first COVID-19 vaccines were approved by the European Union authorities, the roll-out of the vaccination campaign in Germany began promptly and under a legally binding prioritisation [3,4]. With healthcare workers (HCWs) being among the top-priority groups to be inoculated, the logistics and organisation of the vaccination campaign within hospitals were mostly delegated to the hospitals themselves. In the state of Bavaria, a legal framework between the state and the Bavarian Hospital Association was set to define the scope and parameters of the hospitals’ mandate to coordinate the inoculations of their employees [5]. 

As one of the largest hospitals in Germany, the Ludwig Maximilian University (LMU) Hospital was faced with assembling a large-scale vaccination centre within days following the authorisation of the first COVID-19 vaccines by the European Medicines Agency [4]. Due to the rapid setup of the vaccination centre and the lack of opportunity for pilot testing the processes prior to implementation, there were no insights into how well the centre would function. Furthermore, the organisation of a single vaccination centre under the given prerequisites presented a sharp divergence from the established practices at the hospital, where vaccination campaigns (e.g., against influenza) have been routinely set up in a decentralised form with no need for a follow-up visit. Accordingly, it was uncertain how this new form of a vaccination process for the LMU University Hospital would be perceived by its employees and if there were any factors affecting the satisfaction with the process.

Therefore, this paper aims to explore the feasibility of the first-of-its-kind large-scale COVID-19 vaccination centre at the LMU University hospital and assess managerial and implementation aspects that may help facilitate the organisation of in-hospital vaccination centres in similar settings, especially in the context of future outbreak prevention strategies. Further, we analyse the satisfaction of HCWs with the vaccination process and identify potential associated factors that can serve as guidance for the design of vaccination centres for HCWs.

## 2. Materials and Methods

Within the scope of the LMU University Hospital as well as of this analysis, we define all hospital employees, including non-medical hospital staff and medical students, as HCWs. We present a process evaluation consisting of an appraisal of the vaccination centre’s organisation. In addition, we executed 2 online-based anonymous surveys evaluating the HCWs’ satisfaction with the organisation of the vaccinations process. The surveys were part of an extensive evaluation of the whole vaccination campaign at LMU University Hospital within the scope of the prospective study IMPF^LMU^ with the first part of the project exploring the COVID-19 vaccination intent and associated factors in HCWs (1 survey) and the second part, presented here, with 2 surveys, focusing on the implementation of the vaccination centre [6]. The results of the first part of the project have already been published [6]. 

Of the 2 surveys presented here, one was aimed at HCWs who had received at least 1 inoculation (vaccinees) at the in-hospital centre, while the other one targeted HCWs working as staff in the centre. The vaccination centre began operations on 28 December 2020, and remained open until 18 June 2021. This period constitutes the first vaccination campaign against COVID-19 at the LMU University hospital, and is subject to the contents of the following analyses. 

### 2.1. Organisation and Programme Monitoring of the In-Hospital Vaccination Centre

The in-hospital vaccination centre was organised in accordance with the guidelines provided by the Bavarian State Ministry of Health as well as with the recommendations of the Bavarian State Office for Health and Food Safety [7,8]. The centre was set up as a one-way street in a spacious, barrier-free area inside the main hospital building, with separate entrances and exits allowing for an isolated flow for incoming and outgoing vaccinees. There was no intersection with patient care. The space was equipped with a secure network and telephone connection. 

In the admission area located at the entrance, the vaccinees were registered and their COVID-19 vaccination history and recovery status were prompted. Next, the inoculations were given in private cubicles mainly by physicians. Lastly, vaccinees were asked to rest for 15 min in the observation area, located near the emergency equipment and the nearest exit to the emergency department. The vaccinations were prepared mainly by nursing staff and, for hygienic reasons, outside the main area. Opening hours of the centre were from 9 am to 3 pm with occasional extension to 5 pm. We planned for up to 72 vaccinations per hour. 

All HCWs were offered inoculation against COVID-19 in accordance with their professional risk of exposure and with the health authorities’ prioritisation scheme. During the first vaccination campaign, the centre inoculated solely with the Comirnaty^®^ vaccine [9]. The appointment scheduling was arranged using a HTML5 booking system by Mayflower GmbH [10]. For any inquiries or comments concerning the vaccinations, an email address was set up. Daily briefing and on-demand debriefing sessions allowed for continuous adjustments to the workflow of the centre. Numbers of vaccinees and large-scale amendments of the centre´s organisation were discussed with the hospital board on a regular basis. The vaccination centre’s documentation serves as reference for the final data following the end of the campaign. Further details are reported elsewhere [11].

Here, we assess the first COVID-19 vaccination campaign (December 2020–June 2021) at the LMU University hospital, taking into account documentation, observation and emails addressed to the vaccination centre’s inbox. 

### 2.2. Satisfaction with the Vaccination Process

The perception of the vaccination process and centre was evaluated within the scope of the prospective study IMPF^LMU^ [6]. For collecting data on the satisfaction with the vaccination process, 2 questionnaires were created using LimeSurvey Version 4.4.12 + 210308. Most items were measured with a 5-point Likert scale (1 (disagreement/dissatisfaction)–5 (agreement/satisfaction)). Both surveys cover the period of the first vaccination campaign at the hospital (December 2020–June 2021) and were communicated by email as well as via the designated intranet page of the project between 14.04.2021 and 30.06.2021. Participation was voluntary and informed consent was obtained in electronic form. 

Survey 1 was aimed at hospital employees with at least 1 COVID-19 shot at the in-hospital vaccination centre. It assessed satisfaction with the vaccination process and potentially associated factors. The design of the questionnaire was informed by the SAGE Working Group’s guidance on vaccine hesitancy and consisted of 5 sections [12,13]. The questionnaire was communicated via the designated intranet page and was available to the target group between 14.04.2021 and 30.06.2021. The sociodemographic and occupational characteristics of the cohort were tested for associations with the satisfaction with the process as well as with the reported observations of adverse effects following immunization (AEFIs). Statistically significant associations were considered for adjustment in the further analyses. Further, we tested if and how the contextual influences and geographic barriers affected the overall satisfaction with the vaccination process. As a variable for overall satisfaction, we used the 5-point Likert scale item for “The vaccination process at LMU Hospital was generally well organised”. We tested for an association between AEFIs after the first and second vaccination dose and the general satisfaction with the process. Further, we examined whether the individual AEFIs were associated with any of the sociodemographic factors that showed significant association with the general observation of AEFIs after the first and second vaccine inoculation. We examined the attitudes and potential attitude changes towards COVID-19 non-pharmaceutical interventions using the mean values and standard deviation of the answers on the 5-point Likert scale, including Cronbach’s Alpha for reliability testing.

Survey 2 targeted employees that had worked at the in-hospital vaccination centre since 2020. The design of the questionnaire was informed primarily by the evaluation needs of the hospital and consisted of 4 sections: general organisation and perception of the vaccination process, information about the vaccination process and sociodemographic data. The questionnaire was distributed by the vaccination centre’s management team to a mailing list including all persons on the centre’s duty roster. The questionnaire was available between 14.04.2021 and 30.06.2021. Due to the restricted sample size of the vaccination centre staff, we limited the analysis to a descriptive report including the mean values and standard deviation of Likert scale variables as well as Cronbach’s Alpha for reliability testing.

## 3. Results

### 3.1. Programme Monitoring of the In-Hospital Vaccination Centre

#### 3.1.1. Vaccinations, Personnel and No-Show Rates

Between December 2020 and June 2021, we administered 20,250 vaccine doses amongst the 11,005 active and permanent employees, of which 13,790 (68%) were given to female HCWs, consistent with the higher proportion of women among hospital staff. There were no serious incidents or adverse events after immunization. Vasovagalreactions or near-syncopes were the most common incidents, with a frequency of about 1:1000 vaccinations. 

Organisational adjustments were required during the campaign and within the course of continuous resource evaluation, which referred especially to the personnel management, due to an initial overestimation of staff and time needed per inoculation. Within the first days, we recognised that only five instead of ten minutes per inoculation were needed. This allowed for a substantial reduction of the physicians needed for vaccinations from 12 to 6 per 72 scheduled vaccinees per hour. We also switched from a voluntary deployment system to requesting medical staff from individual departments with support from the executive board of the hospital. With growing experience of the staff and decreased need for consultation concerning safety and side effects, we were able to further reduce the inoculation time to 4 minutes, which summed up to 5 vaccinating physicians per 72 scheduled vaccinees/hour, including a 30 min break for each physician. We also found that the consultant who initially used to be permanently on-site as the centre’s manager for emergencies and medical inquiries was needed for occasional telephone consultations only.

Administration was initially covered by eight employees and was also reduced, first to six, and later to five employees, similar to the medical personnel adjustments. The time needed to process a registration for vaccination summed up to four minutes per patient as well. Further, the preparation of the vaccinations equated to 2.5 min per dose and required a total of 3 people daily. Lastly, one additional person acted as on-site operation manager, monitoring and directing the processes, onboarding new employees and coordinating organisational problems and logistics. In total, 14 employees covered the vaccination centre on site. Administration and vaccine preparation were each supervised on demand by a designated person. Other staff needed for work in the vaccinations centre´s environment, such as cleaning, security, IT, logistics and engineering staff, should be taken into account.

The finalised layout of the vaccination centre is presented in Figure 1.

We documented no-show and extra-show rates on a daily basis. We observed a maximum no-show rate of 5.8% for the first shot and 5.2% for the second shot. Further, we aimed to accommodate HCWs with an impromptu inquiry, i.e., without an appointment. This occurred mostly at the beginning or end of a vaccination days’ series, with a maximum of 10.5% extra-shows for the first, and 12.1% for the second vaccination. Ultimately, we performed a mean of 70 vaccinations/hour (range: 49–84/h). In addition, many HCWs presented at the beginning or after their working hours, as well as during their lunch breaks, independently from the time their appointment was scheduled for. As these HCWs would present within the day they had been scheduled, the irregularity was not documented as no-show or extra-show, but, nevertheless, led to unequal distribution of the work load for the staff on site.

#### 3.1.2. Administrative Organisation

The vaccination appointment booking was initially set up with a low-barrier digital environment without special requirements for personal authorisation via login data. This method swiftly proved to be error-prone, thus triggering an adjustment of the system towards booking via personalised login and automatic generation of the second appointment as well as an appointment confirmation via SMS. The follow-up (second) appointment was scheduled in accordance with the recommendations by the German Standing Committee on Vaccination (STIKO) and the Federal Institute for Vaccines and Biomedicines (PEI) [14,15]. The hospital’s employees were continuously informed about any changes or adaptations to the recommendations as well as to the vaccination process via the hospital’s designated intranet page and via newsletter. A detailed evaluation of the communication campaign and tools implemented at the LMU University hospital is published elsewhere [6]. 

The email set up for inquiries received up to 80 messages/day (approx. 0.7% of employees). The questions or messages were medical in around 20% of cases, and organisational, e.g., related to scheduling, in approx. 80% of cases. Initially, medical inquiries referred mainly to safety and expected side effects of the vaccine, while later on, reports of assumed and observed AEFIs as well as questions related to individual diseases, pregnancy, breastfeeding and COVID-19 antibodies dominated. The frequently adapted recommendations regarding the intervals between inoculations as well as between inoculations and SARS-CoV-2 infection were a source of numerous inquiries. The answers to organisational inquiries as well as the administrative work initiating from these inquiries was mainly covered by one person from the hospital’s administration staff while medical inquiries were handled by the centre’s manager as a medical expert. 

### 3.2. Satisfaction with the Vaccination Centre and Process by Vaccinees

Of 11,005 employees, 1662 participated in the survey for vaccinees. Of those, 1035 filled out the questionnaire in full (Table 1). We observed a high satisfaction rate both with the centre as well as with the process—the individual results are presented in below.

The initial testing showed a significant association of age and sex with the reported satisfaction as well as with the observation of AEFIs after the first and second inoculation. Similarly, occupation showed a significant association with satisfaction and reported AEFIs after the first vaccination dose. Variables with significant associations in the initial testing were used for further analyses in the adjusted models.

#### 3.2.1. Satisfaction with the Process and Vaccine-Specific Issues

The four items for general satisfaction with the vaccination process as well as the nine items for the satisfaction with the individual aspects of the vaccination process demonstrated good reliability (Table 2).

The better fitting unadjusted model showed a link between satisfaction with the vaccination process and accessibility of the vaccination centre and waiting time. Vaccinees dissatisfied with the location of the vaccination centre had a 9.542 higher likelihood of perceiving the vaccination process as rather ill-organised. Further, vaccinees who only partially agreed that the vaccination centre was well accessible had a 5.519 higher likelihood of perceiving the vaccination process as partially ill-organised. Similarly, HCWs not willing to travel over 1 h to the vaccination centre had a 9.502 higher likelihood of perceiving the vaccination process as rather ill-organised (Table 2). Vaccinees that reported a shorter waiting time between registration and inoculation were less likely to perceive the vaccination process as ill-organised. The overall duration of the visit to the vaccination centre did not present any significant association with the satisfaction.

#### 3.2.2. Satisfaction with the Provided Information Sources Prior to Inoculation

We measured the satisfaction of participants with the written information provided upon inoculation (Figure 2). All forms of provided written information demonstrated a satisfactory result, with mean values around “4” (“very helpful”). The four items on satisfaction with medical consultation were only available to those participants who reported that they had requested such consultation upon inoculation (*n* = 177). The items for perceived safety and confidence with the vaccination process provided similarly consistent results at the upper end of the Likert scale.

#### 3.2.3. COVID-19 Health Behaviour following COVID-19 Vaccination

We measured the attitudes and attitude changes towards COVID-19 non-pharmaceutical interventions (NPI) after receiving both vaccine doses (Figure 3). All items demonstrated high mean values, i.e., participants were in agreement with the statements. Solely the statement that NPIs should apply in 2022 demonstrated a tendency to the middle.

#### 3.2.4. Observed Adverse Events following Immunization (AEFIs)

There was a weak significant association between experiencing AEFIs after the first inoculation and reporting a lower satisfaction with the process (Table 3). The data showed weak yet significant associations of increasing age and less frequent observation of pain at the injection site and onset of a known migraine within 24 h after the first vaccination. Regarding the second dose, there were more significant associations following the analogous path: pain at the injection site, fatigue, flu-like symptoms, headache, onset of known migraine within 24 h and circulatory weakness demonstrated to be significantly more often observed by younger participants.

### 3.3. Satisfaction of the Vaccination Centre Staff with the Process and Organisation of the Vaccination Campaign

Overall, 74 vaccination centre staff members participated in the survey, with 54 of them filling out the questionnaire in full (Table 1). Here, we also observed a high satisfaction rate.

The satisfaction of the staff was measured with seven items, where the majority presented a consistent mean above 4.50 (Figure 4). The item for information provision during induction was the only one with a lower mean value and a comparably broad standard deviation (4.28 ± 0.97935).

Similarly, the nine items for spatial arrangement and staff management demonstrated comparable consistency, with all items presenting mean values above 4.0 (Figure 4). The lowest mean value referred to the individual’s perception of preparedness in emergency cases (4.19 ± 0.89177).

The eight items on the quality and helpfulness of the information delivered throughout the process also demonstrated consistent mean values above 4.0 (Figure 5). The item with the lowest mean value and broadest standard deviation referred to the written form on data consent (4.07 ± 1.00662).

Further, we asked staff members about their perception of the vaccinees’ knowledgeability regarding COVID-19 vaccines (Figure 5). The five items demonstrated low internal reliability, where the item with the lowest mean value on the Likert scale referred to the perception if vaccinees had questions about the vaccine process prior to inoculation (3.13 ± 0.99140).

The 10 items for satisfaction with the working atmosphere showed consistent mean values (Figure 6). Only the two items comparing the workload at the vaccination centre with that in the regular jobs of staff members demonstrated particularly low mean values, indicating that the workload was neither lower nor higher than that at the regular workplace of staff members (3.56 ± 1.26888, respectively, 2.1852 ± 1.06530).

## 4. Discussion

To our knowledge, this paper provides the first published insights into the organisation and evaluation of a large-scale in-hospital vaccination centre in Germany on the basis of the experience gathered through implementing and operating a COVID-19 vaccination campaign for 11,005 HCWs.

### 4.1. Organisation of the Vaccination Centre—Implementation Considerations

The currently available literature concerned with the topic of organising a COVID-19 vaccination centre is still narrow and mainly concerned with mass vaccination sites for the general population [16,17,18]. As there is a rather limited body of evidence specifically on the organisation of COVID-19 large-scale or mass vaccination centres in hospitals, and specifically for HCWs, our results allow only for a narrow contextual observation.

The number of vaccine doses inoculated in the centre every day corresponds to the rate in the COVID-19 hospital-based or mass vaccination sites [16,19,20]. It should, however, be noted that our centre operated, even in its initial phase, with a rather limited number of personnel for the inoculations compared to the centres described in the literature so far. This is ascribed to the strictly defined dimension of the target group (HCWs vs. population-wide) as well as to the zero-sum nature of the centre’s roster management with physicians consequently being unavailable to provide health care for patients when assigned to the vaccination centre. However, the time needed per vaccinee as well as the time that vaccinees spent in the centre on average compares to the indicators of population-wide COVID-19 and non-COVID-19 mass vaccination sites [16,20]. It is noteworthy that the vaccination centre described deployed a larger number of physicians as vaccination staff compared to public vaccination centres with a larger proportion of nursing staff or medical assistants. This reflects the staff structure of a university hospital. Nevertheless, this instance might have influenced the satisfaction of the HCWs with the vaccination process.

Further considering personnel management, planning and scheduling staff on a voluntary basis assumes a certain degree of predictability and neglects motivation loss over time, as observed at the beginning of the first campaign. We therefore recommend the later adopted option of a planned roster, as this allows for better reliability. Additionally, the arrival of vaccinees at specific times of the day should lead to further adjustments in the personnel planning in the future, in order to cover the bottleneck timeslots more efficiently. Further, as other case studies have pointed out, an onsite manager who continuously monitors and, as appropriate, adapts the workflow, is highly beneficial to the agile management required in the setting [19].

The lessons learned during the first vaccination campaign, especially concerning personnel, facilitated the setup of the second vaccination phase (October–December 2021), to the extent that we were able to offer a mean of 17 influenza shots/hour (range: 3–32 shots), in addition to COVID-19 inoculations, without staff changes. However, no-show rates rose rapidly, to almost 16% (no extra-shows), likely reflecting several factors specific to the second vaccination campaign: as this period coincided with a sharp rise in SARS-CoV-2 incidence, it is possible that many HCWs had to delay their scheduled vaccination due to an infection [21]; further, this period encompassed several adaptations of the vaccination recommendations that might have interfered with one’s eligibility or motivation for receiving a vaccine [22]; lastly, the prioritisation of vaccinations was lifted at the end of the first vaccination campaign, which may have influenced the accessibility to appointments in other vaccination centres preferred by the hospital’s employees due to their temporal or geographical convenience compared to the in-hospital centre [23].

### 4.2. Satisfaction with the Vaccination Process

Our findings indicate a direct association between the accessibility of a vaccine and the satisfaction with the vaccination process. Especially regarding geographical barriers, the results emphasise the need to improve access and reduce physical impediments even among vaccine-receptive populations. Although the Vaccine Hesitancy Matrix observes the contextual influences independently from the vaccination-specific issues, our findings accentuate the benefit of considering those simultaneously, as geographical and temporal barriers may serve as guidance in the design of vaccination programmes [13]. Admitting the possibility of debate on convenience as a factor in hesitancy models, its effect on the satisfaction with the process and, potentially, future willingness to vaccinate, needs to be addressed in order to increase vaccine uptake, even when the respective campaign is organised at the workplace [24,25,26]. This is a particularly relevant aspect in the context of our centre’s organisation, as many HCWs working outside the main campus had to plan for additional travelling time to and from the vaccination centre. Even if our results show that the large majority of HCWs were willing to travel longer than 1 h to receive their inoculation(s), this outcome should only be considered in the context of limited access to vaccines outside of the hospital’s centre during this period of time in the population-wide vaccination campaign. This especially concerns employee groups that were scheduled for vaccination later on in the campaign, e.g., HCWs without direct patient contact, in administrative or other non-medical positions. Furthermore, the observed association between accessibility and satisfaction indicates a potential issue of providing vaccinations in a single centralised centre rather than in a decentralised form covering all locations of the hospital. Looking ahead, and specifically for settings similar to the LMU University hospital, it would be advisable to systematically explore the advantages and disadvantages of a centralised vs. decentralised vaccination supply including the preferences of HCWs. The factors affecting the satisfaction with the COVID-19 vaccination centre can serve as a reference in these future analyses.

In a similar manner, there is a need for consideration of the experience of AEFIs as a factor potentially influencing COVID-19 vaccine-related decisions in the future, as vaccine adaptation and emerging variants of concern may pose the need for further COVID-19 large-scale and mass vaccination campaigns [27]. The factors affecting the satisfaction with the vaccination process indicated by our results need to be taken into account in subsequent research attempts, as these may generally alter the circumstances and arguments in future vaccine-related decision-making processes by HCWs [12]. Specifically, further examination should be focused on whether the AEFI-related experience after a COVID-19 vaccination could affect any future decisions on receiving another COVID-19 inoculation but also vaccinations against other infectious diseases. This constitutes a crucial topic for research, in view of the HCWs’ influential gatekeeping role for vaccine uptake in the general population, e.g., via provider-based interventions [28,29,30]. Additionally, our results show that the AEFI experience following an inoculation significantly affects the satisfaction with the setting where the inoculation has taken place. This potential confounder should be accounted for in future evaluations of vaccination programs and campaigns. The frequency and distribution of AEFIs reported by participants in our survey are consistent with previously reported data in a comparable setting and population [31].

In terms of COVID-19 behaviour, our results present referential data to the first questionnaire of the IMPF^LMU^ project which examined this topic several months prior to the surveys presented here. The overall adherence to the extension of NPI at the hospital confirms the data from the first questionnaire that showed an association between a positive vaccination status and rather agreeing to the extension of the NPIs validity beyond 2021 incl. PCR testing [6]. However, although this aspect was not explicitly examined in the survey, a shift in the attitude towards not extending NPIs’ validity may be hypothesised. This is especially to be considered against the background of the surveys presented here taking place several weeks after the first survey on vaccine hesitancy. Additionally, the questionnaires presented here were available to the target groups in a time period with a higher rate of fully vaccinated HCWs at the LMU University hospital and a lower incidence of SARS-CoV-2 in the general population; hence, participants may have considered the vaccination against COVID-19 as a sufficient preventive measure in the future as well [21].

Further, the results presented here underline the outcomes of the first questionnaire stating that HCWs with a positive vaccination status are less worried about getting infected with SARS-CoV-2 in their personal or professional environment [6].

The overall satisfaction of the vaccination centre staff and the perception of the workload as neither lower nor higher than usual testify to the fidelity of the implementation. Similar to the report of De Micco et al., our results indicate a strong sense of team spirit and commitment by the centre’s staff, hence underlining the role of leadership and personnel management beyond the formative fulfilment of the required tasks [19]. This aspect is particularly crucial for consideration upon implementing a human resource strategy based on planned duty roster, as Hrehova et al. report a relatively higher incidence of self-reported burnout symptoms among HCWs assigned to work at a mass vaccination centre as part of their regular jobs rather than voluntarily [32].

### 4.3. Limitations

Several limitations need to be considered when interpreting the results of this work.

It is uncertain whether the described personnel management can be transferred to other hospitals or settings, especially regarding the number of physicians instead of medical assistants or nursing staff involved, which rather reflects the staff structure of a university hospital than the requirements of vaccination centres. Still, our results provide an indication of the human resources needed for a large-scale vaccination centre, where the majority of the roles may also be assigned to other HCWs with similar qualifications to perform the given tasks, e.g., planning for five qualified medical assistants instead of five physicians to execute the inoculations.

In terms of administrative support, it should be noted that due to the rapid setup, the initial email inquiries were only partially saved, hence we can only provide a general assessment of number and content rather than a detailed analysis.

Both surveys were available to the target groups for approx. 2 months, thus not covering the complete period of the vaccination campaign. Changes in attitude towards COVID-19 vaccines could have potentially been driven by newly distributed information, adaptations of the recommended vaccination scheme, prioritisation or other factors but were not considered in the survey design. Further, the consistently high uniformity of answers to a majority of the questions did not allow for a meaningful and powerful inferential analysis, hence reducing the evaluation to a rather descriptive report. In terms of design it should be noted that the high satisfaction reported by participants may in part be due to an acquiescence bias despite the specific definition of the middle-point in the Likert scale to every item, or other response biases [33]. Further, although our results reflect the evidence on AEFI observation and age, other potentially related factors were beyond the framework of this analysis [34]. Additionally, we need to note the limited response to the survey for vaccinees. As this was the second survey of the IMPF^LMU^ project, it can be hypothesised that the weaker response could be partially owing to a depleted motivation of HCWs to participate in COVID-19 vaccination surveys. To a certain extent, this hypothesis could be broadened to include a general exhaustion with the topic, since COVID-19 was the predominant issue at the LMU University Hospital and beyond during the period of the survey. Further, the voluntary design of the survey certainly accounts for low participation. We acknowledge that a different or an addition dissemination approach might have facilitated the participation in both surveys: an example of such a strategy would have been to promote the surveys on site using posters presenting a QR code and/or link to both surveys.

In terms of representativeness, the age and sex distribution of participants in the survey for vaccinees is similar to the distribution in the whole target population, thus making the results fairly representative for the HCWs inoculated at the in-hospital vaccination centre.

As a long-term observation was outside of the scope of this evaluation, the displayed outcome lacks information on potential fluctuations of the perception of the vaccination centre and process. That is to be considered against the background of limited vaccine availability at the beginning of the campaign and gradually increasing availability subsequently. However, we feel that a follow-up data collection would not have been meaningful due to the fast-paced changes in recommended inoculated vaccines as well in the general pandemic situation that may have implied further confounding factors which one could have not accounted for.

Nevertheless, our work presents valuable insights into the specifics of organising and managing a large-scale in-hospital vaccination centre. As other studies have showed, HCW vaccination campaigns require a tailored yet accessible and agile approach in order to facilitate the uptake of vaccines [35]. A German-wide analysis of in-hospital COVID-19 vaccination has previously highlighted the accessibility to appointments as well as communication as particularly important aspects in designing a vaccination campaign among hospital-based HCWs [36,37]. The comprehensive description of the hospital’s centre as well as the outcomes of its evaluation provide important guidance towards planning, implementing and assessing similar campaigns in comparable settings and contexts.

## 5. Conclusions

Implementing and managing a large-scale in-hospital vaccination campaign requires a specific focus on the geographical and temporal accessibility of the vaccination centre. An agile personnel management is necessary both in terms of the centre staff as well as on the hospital-wide level, as demands may rapidly change and AEFIs may noticeably affect the working ability of vaccinated HCWs, which may, in turn, affect the provision of care to patients. The potential effect of self-reported AEFI experiences following a COVID-19 vaccination on future decisions on vaccination uptake may represent a particularly relevant topic for research. Additionally, self-reported AEFIs following inoculation need to be considered as a confounding variable in the evaluation of the satisfaction with vaccination campaigns and programs, as these appear to significantly affect the perception of the setting where the respective inoculation has taken place. Regardless of the high effectiveness of vaccinations, campaigns among HCWs should still aim to facilitate the adherence to non-pharmaceutical preventive measures such as wearing a mask, testing regularly, and other personal hygiene standards. This is a crucial factor to be considered in occupational health promotion, as our results underline the need for a strategically selected and tactically implemented set of measures that facilitate the achievement of a paramount goal rather than observing and evaluating a single measure. Future research should aim to examine health promotion campaigns in occupational settings as a whole and observe potential interactive and inversely proportional coherences between the adherences to different health promotional activities.

## Figures and Tables

**Figure 1 ijerph-19-16326-f001:**
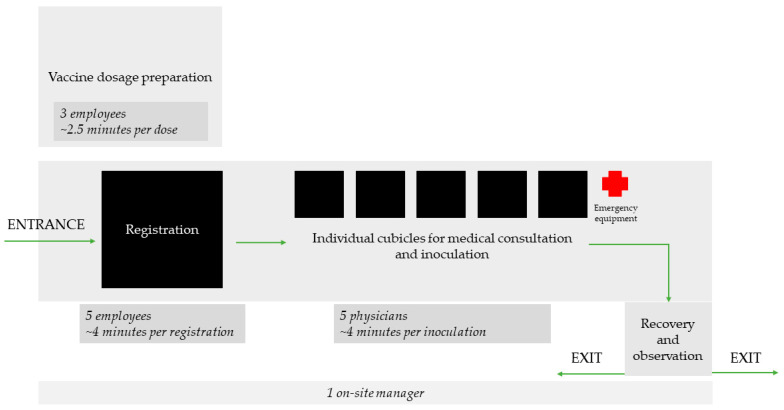
Spatial structure and procedural organisation of the in-hospital vaccination centre of LMU University Hospital after implementing the discussed adaptations to personnel management (December 2020–June 2021).

**Figure 2 ijerph-19-16326-f002:**
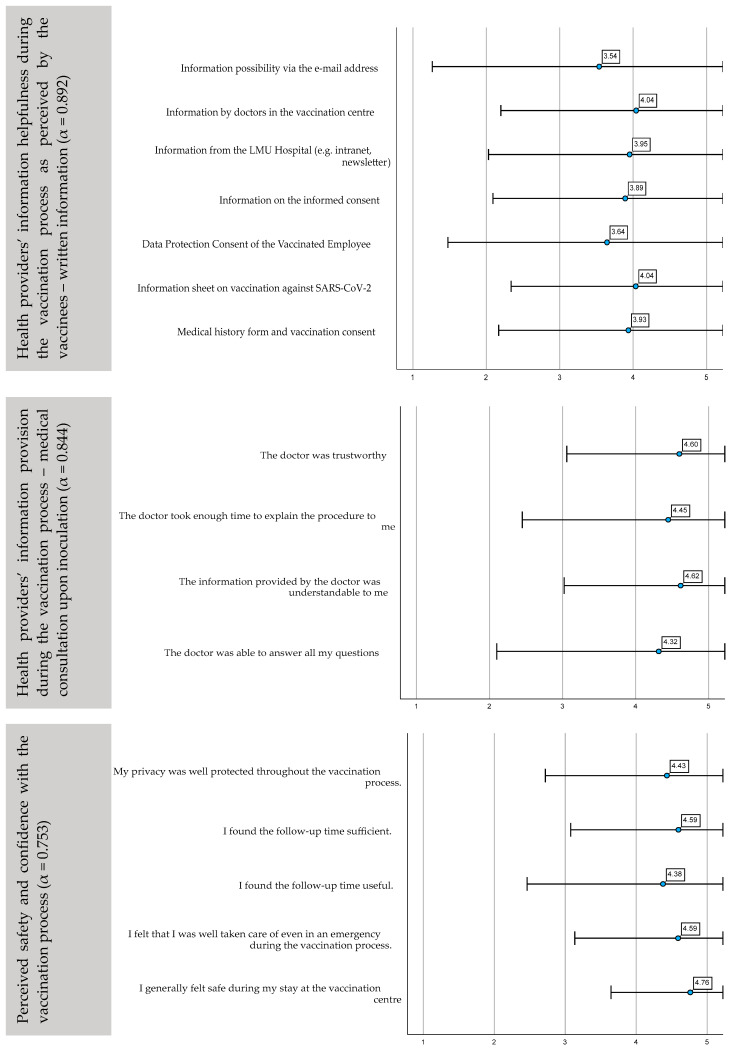
Individual and group influences: health providers’ information quality and helpfulness during the vaccination process as perceived by the vaccinees—written information (α = 0.892), medical consultation upon inoculation (α = 0.844) and perceived safety and confidence with the vaccination process (α = 0.753). Mean values, standard deviation of the answers on the five-point Likert scale and Cronbach’s Alpha.

**Figure 3 ijerph-19-16326-f003:**
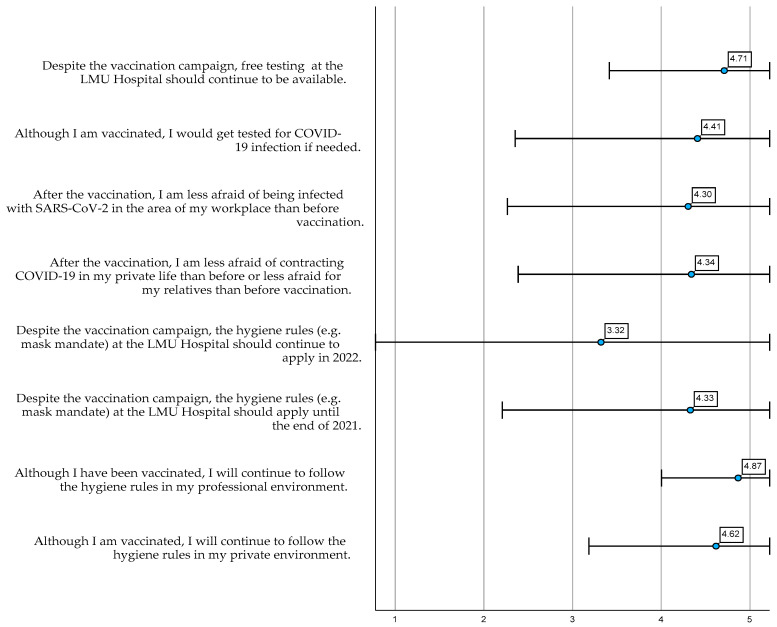
Attitudes and attitude changes towards COVID-19 non-pharmaceutical interventions (α = 0.687). Mean values, standard deviation of the answers on the five-point Likert scale and Cronbach’s Alpha.

**Figure 4 ijerph-19-16326-f004:**
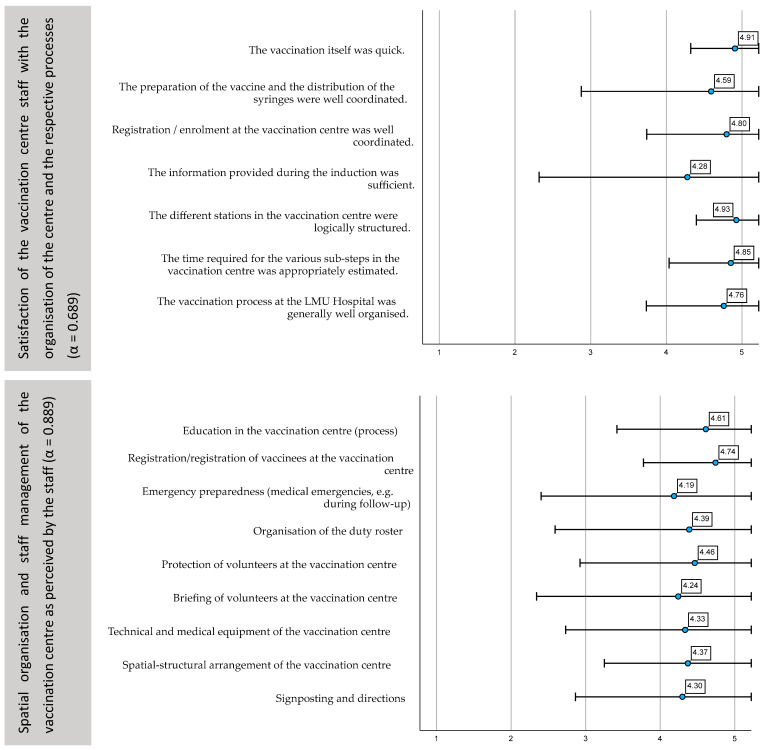
Satisfaction of the vaccination centre staff with the overall organisation of the centre, including spatial-structural layout. Mean values, standard deviation of the answers on the five-point Likert scale and Cronbach’s Alpha.

**Figure 5 ijerph-19-16326-f005:**
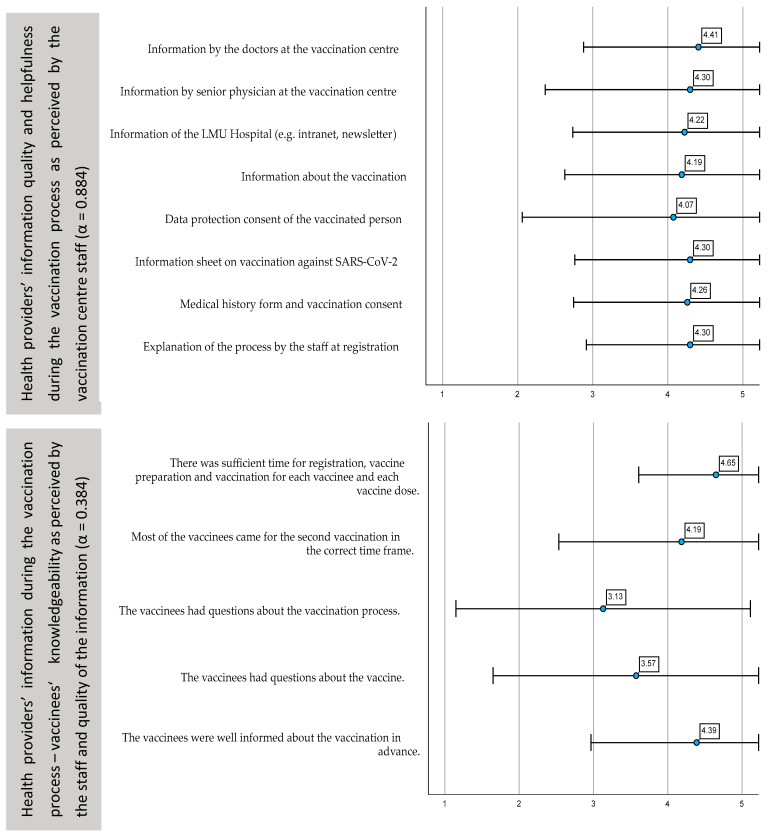
Perception of the vaccination centre staff of the information provision prior to inoculation. Mean values, standard deviation of the answers on the five-point Likert scale and Cronbach’s Alpha.

**Figure 6 ijerph-19-16326-f006:**
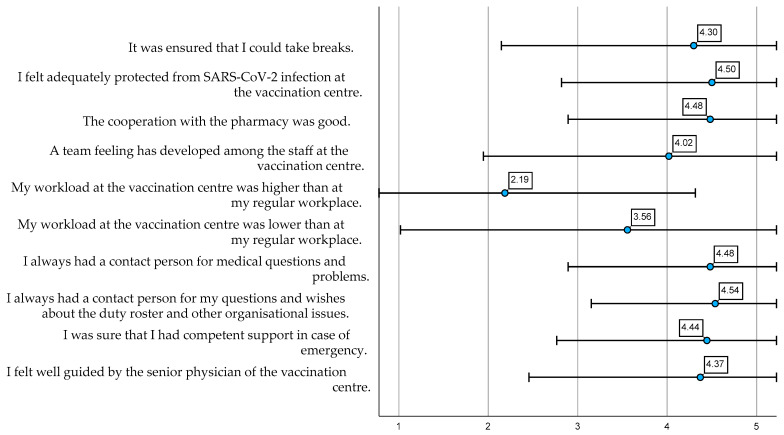
Satisfaction with the working atmosphere at the vaccination centre (α = 0.705). Mean values, standard deviation of the answers on the five-point Likert scale and Cronbach’s Alpha.

**Table 1 ijerph-19-16326-t001:** Sociodemographic and occupational data of surveyed HCWs vaccinated at the in-hospital centre as well as of surveyed vaccination centre staff. Potential factors were tested for association with satisfaction with a chi-square test.

	LMU University Hospital Staff Vaccinated at the In-Hospital Vaccination Centre					Vaccination Centre Staff °	
	*n*	%	Satisfaction with Vaccination Process	AEFIs Following 1st Vaccine	AEFIs Following 2nd Vaccine	*n*	%
Age *			*p* < 0.001	*p* < 0.001	*p* < 0.001		
<29 years	188	14.2				4	7.4
30–39 years	297	22.5				9	16.7
40–59 years	269	20.3				13	24.1
50–69 years	367	27.8				21	38.9
>60 years	189	14.3				7	13.0
No answer	12	0.9				0	-
Sex **			*p* = 0.027	*p* < 0.001	*p* < 0.001		
Male	318	24.1				20	37.0
Female	1001	75.7				34	63.0
Other	3	0.2				0	-
Education			*p* = 0.314	*p* = 0.219	*p* = 0.583		
Secondary/elementary school	31	2.3				1	1.9
Middle school	198	15.0				7	13.0
High school/technical diploma	222	16.8				9	16.7
Vocational training	278	21.0				2	3.7
Academic degree (bachelor)	94	7.1				1	1.9
Academic degree (master’s/diploma)	203	15.4				4	7.4
Academic degree (doctorate or higher)	274	20.7				30	55.6
Other training	21	1.6				0	-
No diploma	1	0.1				0	-
Occupation (dichotomous) ***			*p* = 0.006	*p* = 0.012	*p* = 0.124		
Medical staff	784	59.3				31	57.4
Non-medical staff	538	40.7				23	42.6
Work with COVID-19 patients ****			*p* = 0.916	*p* = 0.123	*p* = 0.699		
Yes	213	16.1				53	98.1
No	1109	83.9				1	1.9
All	1322					54°	

* Age group distribution at LMU University Hospital: <29 years = 22.85%, 30–39 years = 29.11%, 40–59 years = 18.78%, 50–69 years = 20.89%, >60 years = 8.37%. The answer option “No answer” was excluded from the analysis as to not disturb the interpretation of the outcome. ** Sex distribution at LMU University Hospital: Female = 66.3%, Male = 33.7%. *** Occupational distribution at LMU University Hospital: Medical staff = 45.4%, non-medical staff = 54.6%. ° Of the 54 vaccination centre staff members, 47 had signed up voluntarily to support the centre and 7 had been assigned by their department heads; the 54 staff members had the following roles (multiple choice): admission and documentation (*n* = 4), preparation of the vaccination doses (*n* = 13), carrying out the inoculation (*n* = 30), follow-up of the vaccinated employees (*n* = 3), senior physician (*n* = 12), varying role (*n* = 4). **** Mean number of weeks = 23.25 (SD = 22.04, 1–60 weeks). The question was only available to fill out by participants who had selected “yes” to having had worked at a designated COVID-19 unit or with COVID-19 patients. Further, 22 participants answered that they had been working sporadically with COVID-19 patients. In addition to the mean number of weeks, there were 22 participants who reported to have occasionally worked with COVID-19 patients without providing a specific number of weeks. Of the vaccination centre staff, only 1 HCW had worked with COVID-19 patients for a total amount of 4 weeks.

**Table 2 ijerph-19-16326-t002:** Frequency distribution of the satisfaction of LMU Hospital’s employees with the vaccination centre and contextual influences and geographic barriers affecting the overall satisfaction with the vaccination centre tested with multinomial logistic regression (unadjusted model presented, °).

To What Extent Do You Agree with the following Statements? (In Absolute Numbers)					
General satisfaction α = 0.801	Disagree	Rather disagree	Partly agree	Rather agree	Agree
The vaccination process at LMU Hospital was generally well organised.	12	10	64	222	1014
The registration and vaccination process were well organised.	15	27	83	239	958
The different stations in the vaccination centre were logically arranged.	7	8	17	175	1115
The vaccination appointment was easy to organise.	26	38	107	246	905
Satisfaction with the individual aspects of the vaccination process α = 0.808	Disagree	Rather disagree	Partly agree	Rather agree	Agree
Prioritisation of departments to be vaccinated	16	82	203	517	504
Availability of the vaccine	28	160	409	400	325
Organisation of appointment booking	13	53	116	438	702
Scheduling of the administration of the second vaccination dose (availability of appointment options)	9	29	97	328	859
Process of registration at the vaccination centre	5	32	107	417	761
Possibility of a medical consultation at the vaccination centre	7	21	209	352	733
Preparation of the vaccine doses	4	6	222	278	812
Inoculation	5	7	37	266	1007
Follow-up after the inoculation	6	43	306	434	533
	The vaccination process at LMU Hospital was generally well organised. (item used for testing of general satisfaction)				
To what extent do you agree with the following statements?	Disagree/rather disagree		Partly agree		Rather agree/agree (ref.)
Location α = 0.164 AIC = 77.531 BIC = 129.400	*n* (RR; *p*-value)		*n* (RR; *p*-value)		*n*
The vaccination centre at the LMU hospital was easily accessible in terms of location					
Disagree/Rather disagree	8 (9.542; 0.000)		11 (5.519; 0.000)		51
Partly agree	2 (1.478; 0.616)		7 (1.492; 0.339)		111
Rather agree/agree (ref.)	12		46		1074
Even if it had taken me over 1 h to get there to receive the vaccine, I would still have taken the time to get there.					
Disagree/rather disagree	7 (9.502; 0.000)		0		38
Partly agree	1 (1.568; 0.669)		5 (1.775; 0.241)		58
Rather agree/agree (ref.)	14		59		1140
Waiting time * α = 0.706AIC = 94.242 BIC = 166.699	Disagree/rather disagree		Partly agree		Rather agree/agree (ref.)
How long was the waiting time from registration at the vaccination centre until you received the inoculation?	*n* (RR; *p*-value)		*n* (RR; *p*-value)		*n*
Less than 10 min	5 (0.027; 0.000)		34 (0.565; 0.473)		696
Between 10 and 20 min	8 (0.100; 0.006)		20 (0.453; 0.315)		437
Between 20 and 30 min	2 (0.234; 0.112)		6 (0.991; 0.991)		65
Over 30 min (ref.)	6		3		29
I cannot remember	1		1		9
How much time did you spend at the LMU Hospital vaccination centre in total?					
Less than 30 min	8 (0.846; 0.856)		23 (0.507; 0.411)		570
Between 30 and 45 min	7 (0.427; 0.331)		32 (0.806; 0.785)		520
Between 45 and 60 min	1 (0.107; 0.054)		6 (0.437; 0.316)		119
Over 1 h (ref.)	5		3		23
I cannot remember	1		0		4

* For the purposes of this analysis the answer option “I cannot remember” was removed as to not disrupt the statistics. The confidence intervals were removed for better readability. ° Multinomial logistic regression model. The models adjusted for age, sex and occupation did not present a significant association with the satisfaction and were therefore not preferred for the further analyses performed.

**Table 3 ijerph-19-16326-t003:** Adverse events following immunization observed and reported by vaccinees.

Effect of the Observation of AEFIs on the General Satisfaction ^+^							RR, *p*-Value		
Did you observe any adverse reactions after the first vaccination dose?—Yes (*n* = 676)							−0.479, 0.001		
Did you observe any adverse reactions after the second vaccination dose?—Yes (*n* = 924)							−0.052, 0.745		
AEFIs following 1st Vaccine * *n* = 687		Intensity of adverse reaction					Age °	Sex ** °°	Occupation (med vs. non-med) °°
	*n*	Not at all	Very mild	Mild	Strong	Very strong	Kendall Tau *p*-value	Cramér’s V *p*-value	Cramér’s V *p*-value
Pain at the injection site	591	21	96	185	186	103	−0.090 *p* = 0.004	0.153 *p* = 0.008	0.099 *p* = 0.212
Redness	571	415	91	44	14	7	−0.008 *p* = 0.413	*p* = 0.178 °	*p* = 0.163 °
Haematoma	566	509	25	16	12	4	−0.010 *p* = 0.393	*p* = 0.377 °	*p* = 0.689 °
Fatigue	581	193	103	115	91	79	0.013 *p* = 0.347	0.130 *p* = 0.044	0.113 *p* = 0.115
Flu-like symptoms (e.g., aching limbs, chills)	568	392	74	42	25	35	0.021 *p* = 0.283	0.088 *p* = 0.357	0.039 *p* = 0.933
Headache	578	331	79	73	48	47	−0.046 *p* = 0.094	0.105 *p* = 0.179	*p* = 0.540
Known migraine (triggering of an attack within 24 h)	561	533	7	7	4	10	−0.069 *p* = 0.035	*p* = 1.000 °	*p* = 0.878 °
Known tension headache (triggering an attack within 24 h)	556	521	13	8	5	9	−0.015 *p* = 0.350	*p* = 0.297 °	*p* = 0.367 °
Dizziness/balance problems	568	468	45	29	15	11	0.026 *p* = 0.244	*p* = 0.168 °	0.123 *p* = 0.071
Circulatory weakness	567	496	35	18	10	8	−0.033 *p* = 0.192	*p* = 0.246 °	*p* = 0.949 °
Fever ≥ 38 °C	568	518	21	11	8	10	−0.038 *p* = 0.153	*p* = 0.060	*p* = 0.065 °
Nausea, vomiting	569	523	22	14	5	5	0.058 *p* = 0.063	*p* = 0.511 °	*p* = 0.877 °
Diarrhoea	563	525	23	6	6	3	0.056 *p* = 0.068	*p* = 0.789 °	*p* = 0.501 °
AEFIs following 2nd Vaccine * *n* = 935	*n*	Not at all	Very mild	Mild	Strong	Very strong	Age	Sex ** °	-
Pain at the injection site	827	64	217	255	170	121	−0.155 *p* < 0.001	0.142 *p* = 0.002	-
Redness	795	595	115	52	15	18	0.011 *p* = 0.361	*p* = 0.137 °	-
Haematoma	786	714	37	16	11	9	−0.014 *p* = 0.326	*p* = 0.018 °	-
Fatigue	820	97	87	175	216	246	−0.054 *p* = 0.029	0.129 *p* = 0.009	-
Flu-like symptoms (e.g., aching limbs, chills)	796	269	103	103	133	189	−0.090 *p* = 0.001	0.131 *p* = 0.009	-
Headache	807	287	117	141	127	135	−0.091 *p* = 0.001	0.192 *p* < 0.001	-
Known migraine (triggering of an attack within 24 h)	764	710	11	7	14	22	−0.065 *p* = 0.022	*p* = 0.629 °	-
Known tension headache (triggering an attack within 24 h)	769	670	20	14	35	30	0.002 *p* = 0.477	*p* = 0.018 °	-
Dizziness/balance problems	792	582	71	55	51	33	−0.047 *p* = 0.064	0.147 *p* = 0.002	-
Circulatory weakness	781	600	76	50	38	18	−0.050 *p* = 0.055	0.133 *p* = 0.008	-
Fever ≥ 38 °C	778	538	54	57	58	71	−0.064 *p* = 0.018	0.078 *p* = 0.312	-
Nausea, vomiting	794	671	47	35	25	16	0.032 *p* = 0.150	*p* = 0.004 °	-
Diarrhoea	781	705	28	24	14	10	0.035 *p* = 0.134	*p* = 0.448 °	-

^+^ Ordinal regression for an association between AEFIs after the first and second vaccination dose and the general satisfaction with the process. The unadjusted model was preferred for interpretation because in the model adjusted for age, sex and occupation 52.1% of the cells were with zero frequencies. The reference group for both items is “No”. * Excluded from the analysis are adverse events that were additionally added by participants under the category “others”. After the first vaccine, 85 participants reported experiencing “other” AEFIs; after the second vaccine, the reports of “other” adverse events were 143. ** For the purposes of this analysis, we have excluded the item “other” (*n* = 3) as it would not permit the execution of the test. *n* = 1319 (N_missing_ = 3). ° Kendall Tau correlation test. °° Chi-square test (Cramér’s V coefficient). For sex and occupation, the Fisher’s exact test *p*-value is reported where the expected cell count of 20% or more of the cells is lower than 5.

## Data Availability

The presented survey data in this study are available on request from the corresponding author. The data are not publicly available due to the data protection policy of the LMU University Hospital.

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
