# Peer review of "Are We Prepared for the Next Pandemic? Management, Systematic Evaluation and Lessons Learned from an In-Hospital COVID-19 Vaccination Centre for Healthcare Workers"

_ijerph, 2022, doi:10.3390/ijerph192316326_

Round 1
Reviewer 1 Report
Clarity on the research question would be appreciated. There's a lot going on in this paper. On the one hand, you have the set up of the vaccination centre, which is fine and is useful from the perspective of other providers who are interested in this, but then you have two surveys, and it's not clear at the outset what the surveys are measuring or why they are needed. So from the beginning, you should make a greater effort to provide for a clear research question to show how the two parts relate to each other - to the extent that they do.
If I am understanding correctly - this location has many employees but the survey(s) do not have a strong response. This should be addressed more. Are these responses representative of the whole employee population? It does not appear that they are.
They perceive it as ill-organised - so what? Is this a general attitudinal problem? From an interpretation perspective, I might have more questions than answers after reading the results.
The authors are correct that this does provide just a snapshot during a rather long period of response, but the paper deals with an unusual topic, so it is a unique contribution, potentially.
Author Response
Please see the attachment. The file includes the comments and point-by-point responses to both reviewers.

Reviewer 2 Report
Abstract
The objective of this study has been well established in this abstract section that identified the different attributes regarding the covid-vaccination camping for Healthcare workers. Basic challenges have been highlighted for the largest scale along with its urgency. Hence the main focus of this research is to focus on the in-hospital vaccination process. Moreover, this section has also provided a brief idea of research methods and Research results for providing a brief description To The Reader.
Introduction
The introduction section of this study has represented the undertaken preventive measures by the European Union and a brief description of the vaccination campaign in Germany. Moreover, the basic understanding of how Ludwig-Maximilians-University Hospital have faced issues regarding large-scale vaccination strategy has also win depicted in the section for identifying the potential factor of vaccination practices. In addition selection of the top priority groups for the vaccination campaign has also win depicted in the section as well.
Literature review
A clarified literature review section has not been identified in the study however a literary idea has been represented throughout the article. In addition, the basis of managing vaccination strategy has been identified through the study which is related to literary ideas about contextual observations. In spite of having a limited body of literature, the study has depicted a Clear View of how to identify the center of operation at its initial Phase to manage this large-scale vaccination campaign adequately.
Method
The vaccination campaign undertaken by LMU university hospital husband analysed in this paper for non-medical hospital staff and medical students as healthcare workers. Hence the study has Undertaken a quantitative method through the survey. Hence the study has taken approximately 72 vaccinations per hour. Additionally, the professional risks and the privatization schemes have also been demonstrated in the study of how the vaccination process has been done. The study has also focused on the number of satisfaction during this survey and has also been reported based on the target group. The study has taken descriptive analysis based on the values and standard deviation for relativity testing.
Research result
The resort has highlighted that approximately 68% female health care workers are given these vaccinations. Moreover, shifts in the voluntary system have also been highlighted in the research paper with adequate reasoning. Moreover, approximately 13 employees have to contact medical personal adjustments which have also been depicted in this paper adequately. It has been reported that the vaccination operation started on 28 December 2020 and remained until 18th June 2021. Hence the events of immunization have also been observed that approximately 1105 Healthcare workers got vaccination during the campaign.
Conclusion
The conclusion section has provided either about the implementation considerations to identify the different attributes of the vaccination process. Moreover, the limitations of the study have also been depicted in the section adequately.
Comments
The researcher has selected is a specific group for this campaign which can create a lack of transparency in server results regarding the rate of satisfaction. In order to minimise the research gap the researcher can follow a wide range of populations for generating this survey. Moreover, privatization factors can be restructured for developing the survey design which can maximize the transparency of the research outcome.
The authors need to enrich the references
Author Response

(The authors gave the same response as above.)

Round 2
Reviewer 1 Report
The authors have addressed my concerns sufficiently with their revisions. This is an interesting and useful contribution.